# Predicting the Landscape Epidemiology of Foot-and-Mouth Disease in Endemic Regions: An Interpretable Machine Learning Approach

**DOI:** 10.3390/v17101383

**Published:** 2025-10-17

**Authors:** Moh A. Alkhamis, Hamad Abouelhassan, Abdulaziz Alateeqi, Abrar Husain, John M. Humphreys, Jonathan Arzt, Andres M. Perez

**Affiliations:** 1Department of Epidemiology and Biostatistics, Faculty of Public Health, Health Sciences Centre, Kuwait University, Kuwait City 13060, Kuwait; hamoodbohassen@gmail.com (H.A.); h.abrar@ku.edu.kw (A.H.); 2Environmental and Life Sciences Research Center, Kuwait Institute for Scientific Research, Kuwait City 13109, Kuwait; q8vet@hotmail.com; 3Agricultural Research Service, National Bio and Agro-Defense Facility, U.S. Department of Agriculture, Manhattan, KS 66502, USA; john.humphreys@usda.gov (J.M.H.); jonathan.arzt@usda.gov (J.A.); 4Department of Veterinary Population Medicine, College of Veterinary Medicine, University of Minnesota, Saint Paul, MN 55455, USA; aperez@umn.edu

**Keywords:** foot and mouth disease, Middle East and North Africa, interpretable machine learning, ecological niche models, risk-based surveillance, spatial epidemiology

## Abstract

Foot-and-mouth disease (FMD) remains a devastating threat to livestock health and food security in the Middle East and North Africa (MENA), where complex interactions among host, environmental, and anthropogenic factors constitute an optimal endemic landscape for virus circulation. Here, we applied an interpretable machine learning (ML) statistical framework to model the epidemiological landscape of FMD between 2005 and 2025. Furthermore, we compared the ecological niche of serotypes O and A in the MENA region. Our ML algorithms demonstrated high predictive performance (accuracies > 85%) in identifying the geographical extent of high-risk areas, including under-reported regions such as the Southern and Northeastern Arabian Peninsula. Sheep density emerged as the dominant predictor for all FMD outbreaks and serotype O, with significant non-linear relationships with wind, temperature, and human population density. In contrast, serotype A risk was primarily influenced by buffalo density and proximity to roads and cropland. Our in-depth interaction and Shapley value analyses provided fine-scale interpretability by interrogating the threshold effects of each feature in shaping the spatial risk of FMD. Further implementation of our analytical pipeline to guide risk-based surveillance programs and intervention efforts will help reduce the economic and public health impacts of this devastating animal pathogen.

## 1. Introduction

Foot-and-mouth disease (FMD) virus continues to pose a devastating threat to animal and public health, both directly and indirectly. FMD has exerted unprecedented economic and food security consequences on global livestock production systems [1]. This is because the virus can rapidly infect a wide range of domestic and wild cloven-hoofed animal species [2]. Consequently, this leads to severe reductions in food production, disruption of international trade, and the most expensive financial burdens associated with control and prevention resources [3]. In fact, the extensive antigenic and genetic diversity across its serotypes and their corresponding emerging strains complicate vaccine development and necessitates continual revision of immunization strategies [4]. Thus, it was estimated that the annual economic losses in FMD-endemic regions have reached up to $21 billion, while in FMD-free countries, they have reached up to $1.5 billion due to the costs of prevention resources (e.g., vaccines) [5,6]. Unfortunately, the continuous circulation and emergence of novel viral strains in endemic regions, particularly across Africa and Asia, perpetuate cycles of poverty by undermining the resilience of pastoral and smallholder farming systems [3]. Moreover, the disease poses ecological threats by infecting wildlife species, disrupting ecosystem dynamics, and contributing to biodiversity loss [7]. Hence, intervention efforts face significant challenges due to the unparalleled complexity of FMD epidemiology, which is influenced by dynamic host and environmental factors. Subsequently, rigorous interrogation of these factors maintaining population-level FMD persistence and spread is critical for designing targeted surveillance systems and intervention programs. In the past decade, Machine Learning (ML) models have provided profound predictive insights into the spatial epidemiology of FMD on both local and global scales [8,9,10]. Additionally, recent advancements in ML statistical frameworks, particularly interpretable ML algorithms [11], have demonstrated their ability to provide deeper mechanistic insights into epidemiological drivers of complex pathogens like FMD [12].

FMD virus (FMDV, Aphthovirus: Picornaviridae) is a highly contagious RNA Picornavirus that affects cloven-hoofed animals, including cattle, pigs, sheep, camels, and various wildlife species [13]. The virus has seven genetically diverse and immunologically distinct serotypes, including O, A, C, Asia-1, SAT1, SAT2, and SAT3, each with numerous rapidly evolving subtypes [2]. Transmission occurs primarily through direct contact with infected animals via secretions such as saliva, vesicular fluid, milk, and feces. Meanwhile, indirect transmission can occur through contaminated fomites, vehicles, animal products, or aerosols [13]. Furthermore, the virus has shown the capacity to spread over short distances (within 2 km between premises) and long distances (up to 200 km) [14]. Animals with early subclinical infections are highly contagious, while vaccinated or recovered animals subsequently exposed to the virus may become asymptomatic carriers with questionable epidemiological significance [13,15]. Among wild animals, the African buffalo (*Syncerus caffer*), which inhabits Sub-Saharan Africa, is established as the most epidemiologically relevant wild host reservoir for FMDVs [16].

FMD remains endemic in large parts of Africa and Asia. At the same time, countries in Europe, North America, South America, and Oceania have largely eradicated the disease through strict biosecurity, vaccination, and trade controls [17]. However, sporadic incursions still occur, often linked to illegal animal movements or contaminated products [18]. Also, endemic regions have been classified into seven geographical FMDV pools that harbor similar viruses. This classification was mainly based on similarities in ecological and host (i.e., livestock populations and exchange) characteristics and cultural traditions [19]. Within these geographical pools, serotype O is the most widespread and is responsible for most reported outbreaks. Additionally, six out of the seven viral pools almost encompass the entire continents of Asia and Africa. Pools 3 (West Eurasia and Middle East) and 4 (Eastern Africa) primarily cover the Middle East and North African (MENA) regions with frequent incursions from Pool 2 (South Asia). Serotypes O, A, Asia-1, SAT1, and SAT2 are frequently observed in these two pools [17].

In the MENA region, Iraq experienced extensive FMD epidemics during the 20th century, including in 1938, 1945, 1957, and during the 1998–1999 economic sanctions, where over one million cases were estimated [20]. At the same time, countries such as Iran and Turkey served as key reservoirs and dispersal hubs for serotypes O, A, and Asia-1 [21]. Also, the MENA region faced recurring and widespread outbreaks of serotype O in countries such as Algeria (1990, 1999), Morocco (1992), Libya (1994), Tunisia (1994), and Egypt (1997), with periodic involvement of serotype A [22]. Moreover, incursions of SAT2 were also observed, notably in Egypt in 2012, marking its most significant re-emergence since 2006 [23]. Yet, the 2014–2015 emergence of the O/ME-SA/Ind-2001d lineage (which originated from the Indian subcontinent) in Libya, and its westward spread into Algeria, Tunisia, and Morocco, highlighted the ongoing vulnerability of the region to transboundary viral incursions [24]. Nevertheless, the historical spread of the O/ME-SA/Ind-2001a-e lineages from South Asia into the MENA countries has dominated the epidemiological landscape in that region since 2015 and onward [21]. However, in early 2025, severe FMD outbreaks caused by serotype SAT1 (historically restricted to Sub-Saharan Africa) were reported in Iraq, Bahrain, and Kuwait, marking a rare and concerning spread of this serotype into the MENA region [25]. The event also coincides with the emergence and spread of another novel serotype O virus (O/ME-SA/SA-2018), a South Asian lineage, which was detected in Iraq, Iran, and Israel in the first half of 2025.

The MENA region remains the most suitable geographical location for the simultaneous emergence of novel viral strains that actively cross between the endemic viral pool on a global scale [26,27]. In fact, the complexity of FMD epidemiology in that region is rapidly escalating primarily due to unrestricted borders and transboundary livestock movements, ongoing political conflicts, and a weak and heterogeneous veterinary infrastructure. Furthermore, the absence of formal regional surveillance and vaccine-matching programs exacerbates the situation, as vaccines often fail to adequately protect against emerging field strains [28]. The ongoing devastating impacts of FMD outbreaks across MENA countries urgently require a risk-based surveillance program that serves at the forefront of the control and prevention of the disease in the region. However, surveillance systems that depend on traditional analytical methods often fall short because of data heterogeneity, transboundary dynamics, and ecological variability. Nonetheless, in the past decade, the use of machine learning (ML)-based spatial modeling has illustrated the ability to provide deep and novel insights into the complex epidemiology of rapidly evolving pathogens and has substantially grown due to its higher predictive performance [12,29]. Additionally, Machine learning models can handle thousands of heterogeneous variables, are less prone to overfitting, require fewer statistical assumptions, and can effectively quantify non-linear interrelationships in higher-dimensional spaces of the present data [30]. Yet, advances in the ML statistical frameworks, such as interpretable ML algorithms, have not been widely used in modeling the ecological niche of FMD. These algorithms can substantially enhance the interpretation of model predictions and deeply interrogate interrelationships between predictors that shape the risk of outbreaks in their ecosystem [29,30].

In this study, we apply an interpretable multi-algorithm ML statistical framework to publicly available datasets comprising FMD-reported outbreaks in the past two decades and relevant host and environmental risk factors in the MENA region. Hence, we built a spatially implicit predictive risk model for all FMD outbreaks and for serotypes O and A separately. Subsequently, we quantified and compared the predicted spatial probability, feature importance, and interactions that shape the risk of FMD outbreaks for each ecological niche model (i.e., overall niche vs. serotype-specific niche models). Additionally, we aimed to use our spatial predictive models to identify geographical areas mostly suitable for FMD spread and circulation, where significant underreporting issues exist. We also employ a game-theoretic approach, integrated into our statistical framework, that can determine how much each variable in a given ML model contributes to the predicted risk of an FMD outbreak at a specific geographic site, and at the same time. Our rigorous ecological niche modeling approach provides a deeper understanding of the disease’s complex epidemiology across multiple scales. This study contributes a novel, interpretable ML framework for modeling the complex spatial epidemiology of Foot-and-Mouth Disease (FMD) across the Middle East and North Africa (MENA) region. By integrating ecological, climatic, and host-related predictors over two decades of outbreak data, the analysis provides fine-scale insights into serotype-specific ecological niches and the non-linear interactions that drive FMD persistence and spread. These findings offer valuable insights to support the development of risk-based surveillance and intervention strategies in diverse and challenging settings, such as the MENA region.

## 2. Materials and Methods

In this study, we detail a spatially interpretable ML statistical framework based on the analytical approach proposed by Alkhamis et al. 2021 [12], and Fountain-Jones et al. 2019 [31], applied to heterogeneous data sources, and the model training procedures used to analyze Foot-and-Mouth Disease (FMD) outbreaks in endemic geographical areas using the MENA region as an example. Next, we presented the predictive performance of our models, key ecological and host predictors, and the outcomes of global and local interpretability analyses, including partial dependence, interaction strength, and Shapley value assessments. Then, we interpreted these findings in an epidemiological context, comparing serotype-specific risk patterns and highlighting the implications for regional surveillance and control. Finally, we summarized the main insights and outlined the potential applications of this analytical framework for guiding future risk-based intervention strategies.

### 2.1. Outbreak Data

We retrieved all reports of FMD outbreaks in the MENA region between 2005 and 2025 from the Food and Agriculture Organization (FAO) Global Animal Disease Information System (EMPRES-I; https://empres-i.apps.fao.org/ (accessed on 1 June 2025)) and the World Animal Information System (WAHIS) Interfaces (https://wahis.woah.org/ (accessed on 5 June 2025)). The retrieved reports were cross-checked between the two databases, and duplicate or incomplete reports were removed before they were combined. The final outbreak occurrence dataset comprised 2884 geographical locations (i.e., latitude and longitude) reported in 17 MENA countries, including Algeria, Tunisia, Israel, Libya, Iran, Egypt, Turkey, Marocco, Iraq, Kuwait, Sudan, Jordan, Saudi Arabia, Lebanon, Bahrain, Syria, and the United Arab Emirates (Figure 1a). Approximately 76.8% of the outbreaks were caused by serotype O, while serotypes A and SAT2 caused 8.4% caused 5.9% of the outbreaks, respectively; serotype Asia1 caused 1.8%, serotype SAT1 caused 1.3%, and 5.8% of the outbreaks were unidentified (Figure 1b). Most of the affected animal species were cattle, which make up 58.1% of the reported cases. The remaining species included sheep (25.9%), goats (10.6%), buffalo, wild ruminants, or others (3.2%; Figure 1c). We excluded duplicate reports of FMD outbreaks at each unique geographical location to reduce the training error of our ML algorithms. Therefore, we assumed that each location had one or more outbreaks. Thus, the final dataset comprised a total of 1833 (63.6% of all outbreaks) unique FMD-positive sites. We chose to focus our serotype-specific modeling on serotypes O and A, as they were the most prevalent. Additionally, unlike other serotypes, both O and A exhibit a temporal distribution that spans the entire study period, from 2005 to 2025 (Figure 1d), confirming their prolonged endemicity. The final occurrence data for serotypes O and A comprised 1435 (64.8%) and 168 (69.1%) unique geographical sites, respectively. Consequently, we further divided the occurrence data by these two serotypes to model each separately and compare it to the overall risk of FMD in the MENA region.

### 2.2. Host and Environmental Data

For all models, we selected 36 features (i.e., ecological predictors or risk factors) thought to shape the distribution of susceptible host species (e.g., livestock densities), as well as features indirectly linked to their movement (e.g., climate, road density, and land cover). See Appendix A for feature data sources and their attributes. All features were obtained in the form of spatial grids (i.e., raster layers), with different resolutions. 19 bioclimatic variables with a spatial resolution of 5 arc-minutes were obtained from the WorldClim database [32]. These bioclimatic variables, derived from monthly temperature and precipitation data, offer ecologically meaningful representations of climatic conditions. Bioclimatic variables 8, 9, 18, and 19 were excluded from all analyses as they were known to contain spatial artifacts [33]. In addition to the selected bioclimatic variables, we included elevation and wind speed as climatic predictors for FMD outbreaks. We obtained global estimates on the densities of cattle, sheep, goats, and buffalo from the FAO-GeoNetwork database (https://www.fao.org/livestock-systems/global-distributions/ (accessed on 15 October 2025)) with a spatial resolution of 10 min of arc. These livestock densities were estimated based on the observed number of animals per km^2^ adjusted by weights computed by a random forest model [34]. Moreover, we obtained the global ruminant production system grid (GRPS) from the same data source and resolution described above. The GRPS grid is a discrete feature representing the geographical distribution of subsistence-level farming to large-scale operations supplying international markets [35]. These systems are shaped primarily by agro-ecological conditions and land use, though they are also influenced by investment, specialization, and management practices.

Land cover estimates were retrieved from the Global Land Cover–SHARE dataset (Appendix A), with a spatial resolution of 30 arc-seconds. This dataset includes 11 aggregated thematic classes, with each feature representing the proportion of a 1 km grid cell occupied by a specific land cover type. In addition, global satellite-derived normalized difference vegetation index (NDVI) data were obtained from the Copernicus Land Monitoring Service of the European Union, with a spatial resolution of 1 arc-minute (Appendix A). Human population density data were retrieved from the Gridded Population of the World (GPW) database, with a spatial resolution of 30 arc-seconds. These population estimates are based on census and registry data collected between 2005 and 2020 that were harmonized to match administrative boundaries (Center for International Earth Science Information Network—CIESIN—Columbia University, 2018). Finally, a raster with a resolution of 5 arcminutes representing road densities in the MENA region was obtained from the Global Biodiversity Model for Policy Support (GLoBio; https://www.globio.info (accessed on 15 October 2025)) database [36]. This raster integrates data from multiple sources, including OpenStreetMap, in which a United Nations Spatial Data Infrastructure-Transportation (UNSDI-Transportation) Data Model was used to harmonize the individual source datasets.

### 2.3. Data Processing

Our raster processing procedures for the selected ecological features, described above, were implemented using the ‘Raster’ R package (version 3.6-32) [37]. We standardized all selected features onto a single map projection. We then cropped each feature within the spatial extent of the MENA region to limit the subsequent ML analysis to the study area. We also aggregated and resampled our selected features to create a raster stack (i.e., a unified spatial grid) with an approximate spatial resolution of 10 km^2^, as they were obtained from different data sources with varying spatial resolutions. Since our ML pipeline requires presence-absence data, we randomly generated background data point locations that matched the number of observed FMD outbreaks. These generated background data are equivalent to ‘pseudoabsences,’ which is a commonly used approach in classical presence-absence species distribution models [38]. Since FMD is a notifiable pathogen that can spread rapidly across borders, we assumed that the background locations would characterize the ecological variability where FMD outbreaks were not reported, compared to positive locations where outbreaks were observed and reported. We also generated double and triple the number of background points to evaluate the sensitivity of models’ predictions to the proportionality of the generated background points. Thus, the presence-absence outcome data were merged into a single data frame with the selected features.

To evaluate multicollinearity among the selected features, we constructed an intercorrelation matrix and identified pairs of features exhibiting high collinearity (|ρ| > 0.7). Variables with the highest absolute correlations were removed (see Appendix A). Additionally, we used the computational procedures implemented in the ‘Boruta’ R package to further reduce the feature sets to just those relevant for model prediction. This procedure was shown to increase the efficiency and improve predictive performance of ML analytical pipelines, as described elsewhere [39]. We then randomly split the datasets into training and testing sets (80% and 20%, respectively) and subsequently trained the ML algorithms using the K-fold cross-validation approach. We excluded temporal information (i.e., reporting dates of outbreaks) from our models because time-lagged estimates for the selected features were largely unavailable.

### 2.4. Model Training and Evaluation

We developed spatially predictive models for all FMD outbreaks and for each of the two selected serotypes (i.e., O and A). We trained the ML algorithms using the complete feature set listed in Appendix A, excluding features that were eliminated during the data dimensionality reduction procedure described above. We used the analytical pipeline implemented by Alkhamis et al. [12], which simultaneously evaluates and compares the predictive performance of multiple supervised spatially implicit ML algorithms. In this study, we selected the most popular and reliable ML algorithms, including Random Forest (RF), Extreme Gradient Boosting (XGB), Support Vector Machine (SVM), and Logistic Regression (LR), implemented in the ‘Caret’ R package [40]. Together, these algorithms capture a spectrum of model complexity, from linear interpretability (i.e., LR) to advanced non-linear learning (i.e., RF, XGB, and SVM). Specifically, RF is highly robust to noise and overfitting, effectively capturing complex non-linear relationships and interactions among features [41]. While XGB offers superior predictive performance through iterative boosting and regularization, it efficiently detects subtle threshold effects and spatial dependencies [42]. Conversely, SVM performs well with high-dimensional data by finding an optimal hyperplane that maximizes the margin between data classes, resulting in robust decision boundaries and good generalization, even with limited or noisy data [43].

We compared the predictive performance of these ML algorithms because they construct classification models in different ways, potentially leading to varying model outputs and subsequent interpretations. To evaluate the performance of each algorithm, we used a 10-fold cross-validation to infer key performance metrics including the area under the receiver operating characteristic curve (AUC), accuracy (Acc), specificity (Sp), sensitivity (Se), and Matthew’s correlation coefficient (MCC). The 10-fold cross-validation procedure was selected to balance bias and variance in performance estimates, helping to mitigate overfitting and reduce the likelihood of artificially inflated accuracy due to data reuse in both training and validation. Model training was conducted using the default hyperparameter grids for each algorithm. The optimal predictive model for the spatial risk of FMD was determined by comparing the performance metrics across all algorithms using the testing dataset. Yet, to streamline the model selection process of our analytical pipeline, we mainly focused on evaluating the Acc (defined as the proportion of correct predictions out of all predictions), and the MCC values (defined as the measure of the quality of the binary classification). Additionally, spatial sorting bias (SSB) was assessed for each dataset-model combination using the calibrated area under the curve (sAUC), as described elsewhere [44].

### 2.5. Model Interpretation

For each dataset, the algorithm demonstrating the highest predictive performance was used to infer feature importance, feature dependence, and interactions, and examine the relationship between predictors and outcomes at randomly selected individual geographical sites. Given the consistently high predictive performance across models, ensemble methods were not pursued due to increased complexity and reduced interpretability. We used Breiman’s (2001) permutation-based approach to infer feature importance, which is implemented in the ‘iml’ R package [45]. We computed each feature’s global and individual effects on the outcome variable and each observation. To visualize these effects, we generated partial dependence (PD) plots for global effects and individual conditional expectation (c-ICE) plots for observation-level effects, following Goldstein et al. (2015) procedure [46]. Briefly, PD plots illustrate the global (average model) influence of a given feature on the predicted spatial risk, while ICE plots depict the individual-level variation in predictions for each observation, holding all other features constant. We then used Friedman’s H-statistic method, implemented in the ‘hstats’ R package [47], to quantify overall interaction strength for each selected feature in the final models and assessed the top 2-way interactions among features that mainly shape the spatial risk of FMD outbreaks. H-statistic compares the variation in the model’s joint partial dependence (for two features together) to the variation expected if the features acted independently (i.e., it can be interpreted as the percentage of the effect on a prediction that comes from interactions between the features). Lastly, we calculated Shapley values (*ϕ*), derived from game theory, which represent the average contribution of a feature to the prediction across all possible combinations of features. In essence, a feature’s Shapley value indicates how much it increased or decreased the accuracy of the final prediction relative to a scenario where that feature is excluded. Here, we integrated the Shapley values procedure in our analytic pipeline to explore individual-level predictions at randomly selected geographic sites and to assess the contribution of each feature to those predictions [48]. In this final step, we randomly targeted three geographical sites in areas mostly suitable for FMD outbreaks (including serotypes A and O), with predicted spatial probabilities greater than 0.9. At the same time, we selected three sites in areas that are least suitable for FMD outbreaks with predicted spatial probabilities less than 0.1.

## 3. Results

Generally, the highest number of reported outbreaks was in the second half of 2014 and the first half of 2019 (Appendix A), with North African countries such as Algeria and Tunisia reporting the majority of the outbreaks (Appendix A). Yet, our ML models demonstrated remarkable predictive performance, correctly predicting the occurrence of all FMD outbreaks 93% of the time (MCC = 0.85; Table 1). Additionally, both RF and XGB algorithms outperformed other algorithms across all inferred performance metrics (i.e., Acc, Sp, Se, and MCC), with a slight edge for XGB for predicting all FMD outbreaks. However, for serotype O, the Acc and MCC for both RF and XGB were nearly identical, and thus, we resorted to selecting the RF algorithm because it slightly outperformed XGB in terms of Sp and Se estimates. Similarly, the RF algorithms slightly outperformed XGB in predicting outbreaks of serotype A in all model evaluation metrics (Table 1). The predicted geographical range of all FMD outbreaks was nearly identical to that of serotype O (Figure 2b,d). However, the ‘all FMD’ model predicted higher spatial probabilities in the northwestern MENA region than what was inferred by the serotype O model. Most of the predicted outbreak risk areas (*p* > 0.5) remained within the spatial extent of all observed outbreaks (Figure 2b,d,f), but with few notable exceptions. For example, the ‘all FMD’ and serotype A models predicted a wide geographical risk area extending from the west to the East of Syria, despite being the least reporting country (Figure 2a,b,f). Similarly, all of the ML models consistently predicted high spatial risk areas (*p*s > 0.5) in under-reporting countries, including central Saudi Arabia, northern UAE, central Sudan, western Jordan, as well as across Lebanon and Bahrain (Figure 1a and Figure 2b,f). Moreover, countries that did not report any outbreaks, such as Qatar and Yemen, were also predicted as suitable geographical sites for spreading and circulating FMD outbreaks. Despite being trained on the smallest dataset (n = 168 outbreaks), the serotype A model demonstrated mostly acceptable performance metrics, with Acc, Sp, and Se greater than or equal to 0.83 (Table 1). Moreover, it was able to identify newly distinct suitable geographical areas (*p* > 0.75) in southwestern Saudi Arabia, western Yemen, Northern UAE, and Oman (Figure 2f). In fact, our inferred sAUC values for all of the ML models were closer to 1 than 0 (sAUCs ≥ 0.98), indicating significantly weak spatial sorting bias in the retrieved outbreak data (Table 1).

Our ML statistical framework inferred sheep density followed by temperature seasonality as the most important features associated with the predicted spatial risk of all FMD outbreaks in the MENA region (Figure 3a). Temperature seasonality is derived from the standard deviation of monthly mean temperature multiplied by 100. Low values of temperature seasonality represent relatively constant temperatures throughout the year (e.g., at the equator) and high values represent large intra-annual variation in temperatures (e.g., continental climates with hot summers and cold winters). PD plots showed that the spatial risk of all FMD outbreaks showed increased FMD risk even at very low sheep density, but this risk plateaued at values > 500 sheep per 5 km^2^ (Figure 3b). Also, PD plots illustrated that the risk of all FMD outbreaks sharply increased and plateaued when temperature seasonality was at values ≅ 500 (Figure 3c). Wind speed followed by temperature seasonality had the strongest overall interactions with other features (Figure 4a). However, the interaction between sheep density and human population density was the strongest among all other 2-way interactions that shape the risk of all FMD outbreaks (Figure 4b). Our model showed that the spatial risk was the highest when sheep and human densities were >500 animals and >6000 humans per 10 km^2^, respectively (Figure 4c). In addition, Shapley values (Figure 5a) showed that the randomly chosen site in Western Syria suggested a probable FMD outbreak was likely observed there, due to high sheep density (>15,000 animals per 10 km^2^) and wind speeds averaging over 3.3 ms^−1^. Conversely, the site that we selected in southern Saudi Arabia was likely negative (i.e., no outbreak recorded) due to substantially small densities of sheep and human densities (<141 animals per 10 km^2^ and <2 per 10 km^2^, respectively; Figure 5b).

The serotype O model, similar to the ‘all FMD’ outbreak model, showed that sheep density was the most important predictor influencing the spatial risk of corresponding outbreaks. However, wind speed emerged as the second most significant predictor, surpassing temperature seasonality (Figure 2a). Here, we inferred a marked increase in the risk of observing serotype O outbreaks in locations where a threshold of over 500 sheep is observed per 10 km^2^ (Figure 3e) and wind speed is above 3 ms^−1^ (Figure 3f). Also, sheep density followed by wind speed had the strongest overall interactions with other features (Figure 4d). Nevertheless, the interaction between Mean Diurnal Range (MDR) and sheep density was the most influential in shaping the risk of serotype O outbreaks. MDR is the average difference between daily maximum and minimum temperatures across months of the year. High values suggest climates with large day–night temperature swings (e.g., deserts). Low values represent climates with moderated day–night differences (e.g., areas near the ocean). Our model suggested that geographical areas with MDR between 10 and 12.5 °C and sheep densities over 20,000 animals per 10 km^2^ were the most likely to experience serotype O outbreaks (*p* > 0.5; Figure 4f). Additionally, our results illustrate that a serotype O outbreak was likely observed in Northern Libya due to the high sheep density and moderate MDR (≅8000 animals per 10 km^2^ and ≅12 °C, respectively; Figure 5c). In contrast, a serotype O outbreak was likely absent in Eastern Sudan due to low sheep density (≅200 animals per 10 km^2^), temperature seasonality of 420 and MDR of 14.5 °C (Figure 5d).

In contrast to the results of the models described above, the serotype A ML model inferred that buffalo density, followed by road density, were the most important predictors of the corresponding FMD outbreak (Figure 3g). Risk of a serotype O outbreak increased rapidly at low buffalo density (>100 per 10 km^2^) as well as low road density (>100 per 10 km^2^) before plateauing (Figure 3h,i). Further, human population density, followed by buffalo density, had the strongest overall interactions with other predictors (Figure 4g). However, the strongest 2-way interaction was between proximity to cropland and population density (Figure 4h). Here, we inferred that geographical locations with cropland coverage of 10% threshold and above with high human population density (<20,000 per 10 km^2^) were critical for shaping the risk of serotype A outbreaks (Figure 4i). Yet, a serotype A outbreak was likely observed in Northern Egypt due to moderate to low human, buffalo and road densities (1076, 101, and 178 per 10 km^2^, respectively; Figure 5e). In contrast, a site in Southern Algeria was likely negative for serotype A outbreaks due to substantially low human density and absence of cropland (2 per 10 km^2^, 0%, respectively; Figure 5f).

## 4. Discussion

We use an interpretable ML statistical framework on 20 years of outbreak data to untangle deeper insights into the complex spatial epidemiology of FMD and the distinct contributions of host and ecological features to outbreak risk across the MENA region. Our analyses revealed that different host densities and anthropologically related features in combination with temperature variations were the most important predictors of FMD risk. However, our rigorous integration of non-linear interactions between selected features revealed their key role in shaping the risk of FMD in the region. These complex interdependencies provided novel epidemiological insights into how environmental and host factors jointly shape FMD distribution across diverse populations and geographical regions. Markedly, the interaction between host, climate, and landcover features varied between serotypes O and A, indicating subtype-specific ecological dynamics. The models demonstrated high predictive accuracy despite the absence of explicit temporal information, suggesting that our selected ecological features are sufficient to capture spatial risk patterns over smaller timescales (i.e., a few years or even months). These insights not only inform risk-based surveillance efforts in the MENA region but also contribute to reducing the economic consequences of this devastating animal pathogen.

### 4.1. Key Drivers of FMD in the MENA Region

Our ML models revealed that over the past twenty years, host densities and climatic features (especially temperature and wind) have been key elements in establishing a suitable ecosystem for the circulation and persistence of all FMD serotypes in the MENA region (Figure 3 and Figure 4). Moreover, our selected ecological features may have significantly contributed to the continuous emergence of novel strains. Indeed, the specific combination of host densities and environmental features that shaped outbreak risk was notably serotype-specific when comparing the ML inferences between serotypes O and A. Additionally, land cover features and other epidemiological characteristics are likely to differ by serotype. Nevertheless, a deeper comparative investigation of the epidemiological characteristics (i.e., beyond conventional statistics) of different FMD serotypes in endemic geographical locations like the MENA region has not been widely carried out. Yet, Carvalho et al. [49], uniquely compared the epidemiological dynamics between serotypes A and O using a rigorous phylodynamic model that integrated spatial, temporal, environmental, and genetic data of viruses isolated across the South American continent. They found that the spatiotemporal dynamics and evolutionary characteristics between the two serotypes were remarkably distinct. At the same time, the trade of live cattle was the most important predictor for serotype O evolution and spread. Therefore, for rapidly evolving pathogens with multiple antigenically distinct variants, subtype or serotype-specific epidemiological investigation may be more appropriate and taxonomically meaningful [50]. This has been showcased by Alkhamis et al., who implemented a similar serotype-specific ML analytical pipeline on Bluetongue virus (BTV) outbreaks in Europe and illustrated that different BTVs exhibited remarkably different epidemiological and ecological characteristics [12].

Nevertheless, our models did not consider the impact of intervention campaigns, including vaccination, culling, and movement restrictions, since obtaining this information from countries in the MENA region is difficult due to various factors, as mentioned above. Nonetheless, the role of environmental factors in predicting the dynamics of FMD spread is critical for properly assessing intervention campaign effectiveness [51,52]. Thus, the predicted outbreak spatial extent from our serotype-specific models could be used to set risk-based intervention zones. Future research should prioritize evaluating the role of different intervention strategies, particularly vaccination, in modulating the outbreak risk associated with highly prevalent serotypes in different ecosystems (or epidemiological settings) [53].

### 4.2. Role of Host Species and Anthropogenic Factors in Shaping the Spatial Risk of FMD

Many past studies have highlighted the critical role of cattle population dynamics in the spread and maintenance of FMD in different geographical regions [1,4,52]. Moreover, most of the positive cases in our study were recorded in cattle (approximately 58%; Figure 1c). It is worth noting that the detection rate of FMD in cattle is higher than in small ruminants because the clinical signs of the disease in cattle are usually more severe, while in small ruminants, the signs may be mild or absent [54]. Therefore, past studies typically infer cattle population and associated anthropological dynamics as the most significant predictors of FMD risk [55]. Conversely, our study revealed that sheep density had a remarkably higher and more complex role than other species in shaping FMD outbreak risk in the MENA region, particularly for serotype O (Figure 3a–f, Figure 4a–f and Figure 5a–d). This is not surprising because, in the MENA region, livestock systems, particularly pastoral and transhumant systems, often include high-density sheep flocks that mix and travel with other species [56]. Also, besides the fact that FMD pathogenesis differs between sheep and cattle [53], sheep are frequently unvaccinated and typically have subclinical infections, making them effective reservoirs. In addition, sheep-mediated cross-species transmission pathways were confirmed, particularly in Asia [57]. This notion indicates that geographical areas with either moderate or high sheep densities can comprise suitable ecosystems for the circulation and persistence of endemic FMD strains (Figure 3b,e and Figure 4f).

Unlike previous studies, our ML models revealed that outbreak risk is not linearly related to sheep density, but rather strongly interacts with other factors such as temperature seasonality (Figure 3a,c) and human population densities (Figure 3c and Figure 4b). These results should motivate the intensification of surveillance resources during seasonal religious festivities such as Eid Al-Adhaa (which goes back over 1500 years), when livestock (particularly sheep) and human movements are at their highest throughout the year. Such seasonal anthropological dynamics were consistently implicated in the transmission and spread of the FMD across the borders of Islamic majority countries [58]. Furthermore, we inferred wind speed as the second most important predictor for serotype O outbreaks (Figure 3f), while having the strongest overall interaction strength with other features in shaping the risk of all FMD outbreaks (Figure 4a). The Serotype O model illustrates that locations with low and constant wind speeds averaging around 3 ms^−1^ may constitute suitable environments for FMD outbreaks. Past studies hypothesized that gentle wind may facilitate the airborne transmission of FMD by allowing viral aerosols to remain concentrated and stable to travel short to moderate distances. In contrast, stronger, high-speed winds may widely disperse these aerosols (and possibly demolish the carried virus particles), thereby reducing the viral concentration and the likelihood of infection [59,60]. Additional factors such as temperature, humidity, precipitation, and land cover characteristics may influence the distance and direction of spread [14], which our models also captured as important predictors or having significant interactions with other features (Figure 3, Figure 4 and Figure 5). However, it is impossible to rule out fomites and other anthropogenic factors associated with these findings.

### 4.3. Interpretation of Serotype-Specific Ecological Niches

Our serotype-specific ML model revealed that serotype A has a remarkably distinct ecological niche compared to serotype O (Figure 3g–i and Figure 4g–i). Additionally, despite the relatively low number of reported outbreaks (compared to serotype O), we inferred a significantly higher spatial risk in certain geographical locations within underreporting countries, such as Saudi Arabia, Qatar, the UAE, Oman, Yemen, and Syria (Figure 2f). Also, our results illustrates that buffalo and road density were the most important predictors, with low threshold values sufficient for the occurrence of serotype A outbreaks (Figure 3g–i). Unlike SAT1-3 viruses (commonly circulating in Sub-Saharan Africa), serotype A is not typically associated with outbreaks in buffalo populations. However, a newly emerging serotype A strain caused severe outbreaks in buffalo populations in Egypt in 2022, which subsequently became endemic [61]. Small buffalo holdings, situated close to cropland and general roads, are highly prevalent across the Nile region of Egypt. Thus, our model suggests that geographical locations associated with similar ecological features may provide suitable environments for the incursion and circulation of serotype A viruses (Figure 4g–i). Consequently, restricting buffalo movements and mixing with other livestock, as well as intensifying serotype-specific surveillance activities on their populations during emerging FMD epidemics, could substantially help improve control and prevention efforts.

### 4.4. Limitations

Despite the remarkable robustness of our ML models (Table 1), the heterogeneous nature of the outbreak data (i.e., testing and reporting procedures conducted by different countries) and its completeness may pose a significant limitation for study inferences. This may also include underreporting, especially in countries with weak veterinary infrastructure, which often neglect surveillance of FMD in apparently healthy livestock populations. That said, our predictions were epidemiologically plausible in explaining the complex dynamics of FMD across the MENA region, which aligns with the published literature. Additionally, our model was able to rigorously quantify the important role of sheep in the spread and circulation of FMD, even though most of the reported cases were in cattle populations. Another limitation of the present study was that we could not incorporate features that directly represent animal movements within and between affected countries in the region. However, using features such as roads, human population densities, and land cover characteristics (e.g., proximity to crop lands; see Appendix A) as proxies for animal movement not only improved our model’s predictive accuracy but also showed significant statistical importance in predicting the spatial risk of FMD outbreaks.

Future studies should consider true absences when available, as using artificial background points may impact model prediction. Yet, obtaining such information is notoriously difficult due to the geopolitical nature of the MENA region and its surveillance infrastructure. That said, inferences from our final models were not affected by the number of assigned background data points (i.e., no significant changes were observed in the models’ predictive performance or spatial prediction range). Additionally, simplifying the model selection process should be considered to make our proposed analytical pipeline for surveillance. This can be achieved using analytical procedures implemented in the ‘Tidysdm’ framework proposed by Leonardi et al., 2024 [62], which averages predictions from multiple ML algorithms instead of relying solely on the best-performing model for interpretation. However, this approach needs further development, as we still cannot extract individual-level inferences for two-way interactions and Shapley values.

### 4.5. Implications for Risk-Based Interventions

The escalating complexity of FMD epidemiology, coupled with the growth in the size of related data and highly non-linear host and ecological relationships, highlights the strengths of our ML statistical framework. Our data processing workflow in this study effectively transformed heterogeneous and spatially inconsistent raw datasets into a harmonized and analytically robust framework. While the rigorous collinearity assessment and K-fold cross-validation procedure ensured reliable feature selection, model stability, and reproducibility, thereby enabling accurate, interpretable, and spatially explicit modeling of FMD risk across diverse ecological settings. Moreover, we illustrated how our selected ML algorithms outperform simpler and commonly used algorithms, such as logistic regression (Table 1) and maximum entropy models [63]. Furthermore, we illustrated how our statistical pipeline can take advantage of tools like Shapley or c-ICE plots to provide fine-scaled model interpretability. This intuitive attribute can further refine decisions related to intervention activities. For example, in a randomly selected site in Northern Egypt (Figure 5e), specific thresholds of human population, buffalo, and road densities made that site at high risk for a serotype A incursion. In contrast, the other selected site in Southern Algeria (Figure 5f) completely lacks such a threshold, making it unsuitable for the spread and maintenance of serotype A. It is worth noting that we explored and excluded the results of global Shapely analysis as it yielded similar overall results to the feature importance analysis (see Figure 3a,d,g). This is because, in epidemiological terms, global Shapley values reveal the overall importance and magnitude of key features across the entire study area. In contrast, individual Shapley values expose the local ecological dynamics by showing where and why certain regions emerge as disease hotspots while others remain unaffected.

## 5. Conclusions

This study applied an interpretable and adaptable ML framework capable of handling large, heterogeneous datasets to predict the landscape epidemiology (i.e., ecological niche) of FMD in the MENA region. The models consistently identified areas at elevated risk for the emergence, spread, and circulation of all FMD serotypes and effectively distinguished between the spatial dynamics of serotypes O and A. The resulting spatial risk maps revealed new areas for FMD potential virus circulation in countries with significant underreporting. The analysis revealed the important influence of susceptible hosts, climate, and anthropogenic features in shaping FMD outbreak risks, capturing their complex non-linear interrelationships. Furthermore, the study delineated epidemiological characteristics of high-risk regions on finer spatial and serotype-specific scales. The continued application of this analytical pipeline can support targeted interventions, helping mitigate the economic and public health impact of FMD outbreaks in the MENA region and serving as a transferable model for other regions burdened by this devastating animal pathogen.

## Figures and Tables

**Figure 1 viruses-17-01383-f001:**
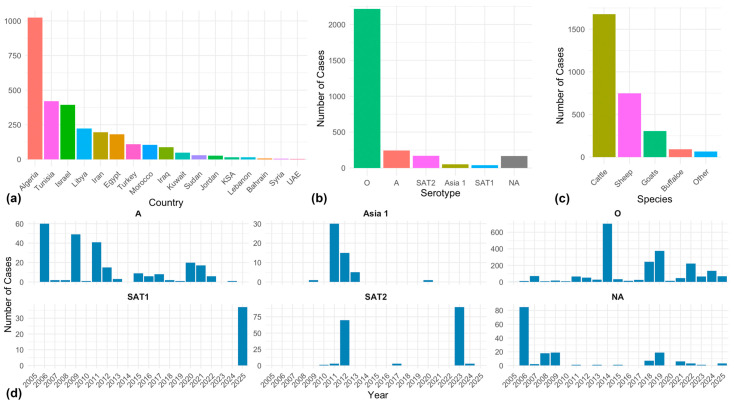
Graphical summary profile of the Foot-and-Mouth Disease outbreaks in the Middle East and Northern Africa between 2005 and 2025. (**a**) Frequency of the outbreaks per country. (**b**) Frequency of each serotype. (**c**) Frequency of the outbreaks per host species. (**d**) Temporal distribution of the outbreaks for each serotype.

**Figure 2 viruses-17-01383-f002:**
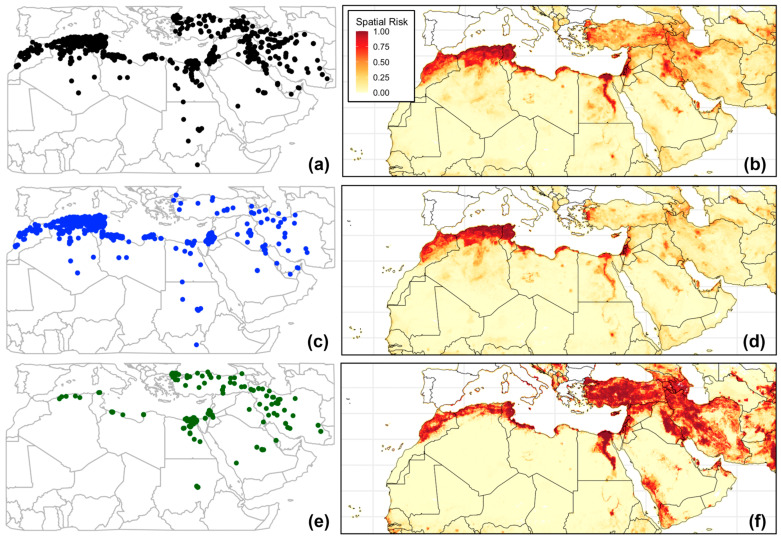
Locations of Foot-and-Mouth Disease (FMD) outbreaks reported in Middle East and Northern Africa between 2005 and 2025. (**a**) Geographical distribution of all FMD outbreaks. (**b**) Predicted spatial risk (*p*) of all outbreaks using extreme gradient boosting algorithm. *p* ≥ 0.75 indicates high risk, 0.75 < *p* ≤ 0.25 moderate risk, and *p* < 0.25 indicates low risk areas (**c**) Geographical distribution of serotype O outbreaks. (**d**) Predicted spatial risk of serotype O outbreaks using random forest algorithm. (**e**) Geographical distribution of serotype A outbreaks. (**f**) Predicted spatial risk of serotype A outbreaks using the random forest algorithm.

**Figure 3 viruses-17-01383-f003:**
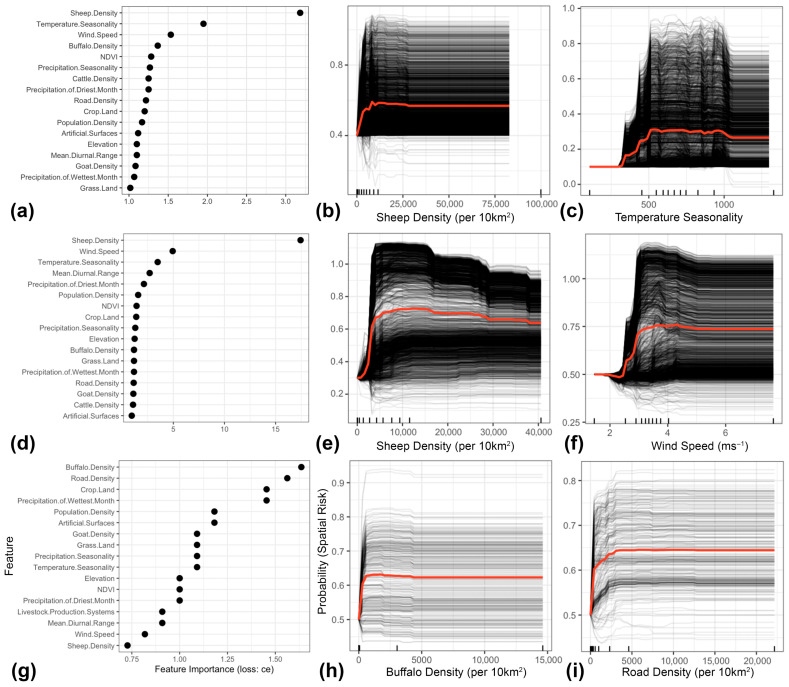
Feature importance plots rank ordering risk factors that contribute to Foot-and-Mouth Disease (FMD) spatial risk in the Middle East (**a**,**d**,**g**). Relative importance was estimated using a classification error loss function (loss: ce). (**b**,**c**,**e**,**f**,**h**,**i**) Illustrate centered individual conditional expectation (ICE) plots for the top two important features that contribute to the risk of FMD. The black lines represent the predicted risk of a FMD outbreak in a specific geographical location, while the red line signifies the partial dependence, calculated as the average risk across all locations. (**a**–**c**) Indicate all FMD outbreaks. (**d**–**f**) Indicate serotype O. (**g**–**i**) Indicate serotype A. Temperature seasonality had no specific unit and calculated as the standard deviation of monthly temperatures (°C) scaled by 100.

**Figure 4 viruses-17-01383-f004:**
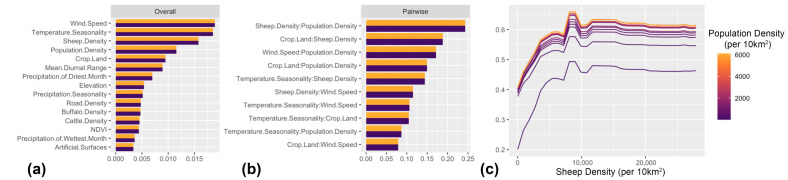
Freidman’s H-statistic feature interaction plots between features that shape the spatial risk of Foot-and-Mouth Disease (FMD) in the Middle East and Northern Africa. (**a**–**c**) all FMD outbreaks. (**d**–**f**) Serotype O outbreaks. (**g**–**i**) Serotype A outbreaks. (**a**,**d**,**g**) Showing overall interaction strength per feature. (**b**,**e**,**h**) Showing the strongest pairwise interactions between features. (**c**,**f**,**i**) Partial dependence plots for the top individual pairwise interaction between features corresponding to the figure on the left.

**Figure 5 viruses-17-01383-f005:**
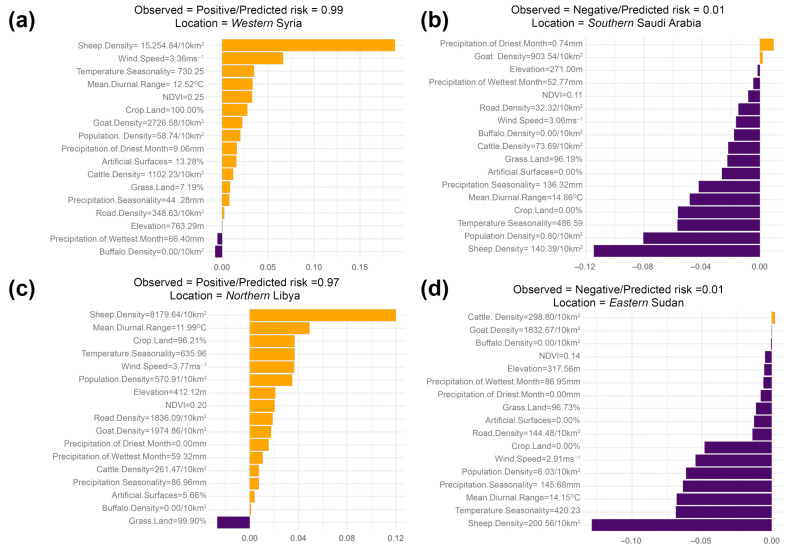
Shapley plots showing feature contributions for each Foot-and-Mouth Disease (FMD) outbreak reported in six randomly selected geographical locations across the Middle East and Northern Africa. Features’ magnitude of contribution was inferred using Shapley values (ϕ). Positive *ϕ* values indicate that this feature increased the risk of a given FMD outbreak. Negative *ϕ* values indicate that this feature decreased the risk of a given FMD outbreak. (**a**,**b**) All FMD serotypes. (**c**,**d**) Serotype O. (**e**,**f**) Serotype A. (**a**,**c**,**e**) Represent positive sites. (**b**,**d**,**f**) Represent negative sites.

**Table 1 viruses-17-01383-t001:** Cross-validation summary results for the performance parameters of the selected machine learning algorithms. Best performing machine learning models are highlighted in gray.

Model	Accuracy (%)	Specificity (%)	Sensitivity (%)	MCC	sAUC
	All Serotypes				
RF	91.74	91.20	92.28	0.83	0.99
XGB	93.26	90.81	93.70	0.85	0.99
SVM	89.80	88.04	91.55	0.79	0.97
LR	87.20	83.71	90.68	0.74	0.94
	Serotype O				
RF	94.94	95.44	94.43	0.89	0.99
XGB	94.95	94.90	93.01	0.89	0.99
SVM	94.19	93.84	94.55	0.88	0.97
LR	91.34	92.75	89.93	0.82	0.95
	Serotype A				
RF	86.07	83.04	84.10	0.72	0.98
XGB	84.29	82.99	82.99	0.71	0.96
SVM	80.87	76.59	85.15	0.61	0.94
LR	82.68	80.89	84.48	0.65	0.93

RF: Random Forest, XGB: Extreme Gradient Boosting, SVM: Support Vector Machine, LR: Logistic Regression, MCC: Mathew’s correlation coefficient. sAUC: AUC for spatial sorting bias. Model highlighted in gray was the best performing model.

## Data Availability

R codes used to generate the results of the present study are available on the following GitHub repository: https://github.com/maalkhamis/Modeling-the-Ecological-Niche-of-Foot-and-Mouth-Disease-in-MENA-Region (accessed on 15 August 2025).

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
