# Peer review of "Predicting the Landscape Epidemiology of Foot-and-Mouth Disease in Endemic Regions: An Interpretable Machine Learning Approach"

_viruses, 2025, doi:10.3390/v17101383_

Round 1

Reviewer 1 Report

Comments and Suggestions for Authors

The manuscript by Dr. Alkhamis and colleagues deals with a machine learning approach to associate foot-and-mouth disease cases with known environmental and eco-epidemiological drivers. The topic of this manuscript is quite interesting, due to the widespread impacts of foot-and-mouth disease in large geographical regions, especially in North Africa and the Middle East. The manuscript is well written and well organized. I only have a couple of technical points on the approach, and some minor comments -- all detailed below.

Technical comments
- I understand, in principle, the inclusion of multiple metrics to evaluate the performances of the machine learning algorithms. However, when it comes to selecting the best performing models, the presence of alternative metrics makes the choice more complicated. For instance, for the `All Serotypes' model, XGB is identified as the best candidate, yet it is outperformed by RF as far as specificity is concerned. As for the `Serotype O' model, RF is selected but XGB outperforms it in terms of accuracy. Trade-offs like these are to be somehow expected in the presence of multiple evaluation metrics, but what is the criterion used for model selection is never formally discussed here. Personally, I would recommend to pick just one metric to conduct the selection or -- perhaps even better -- to adopt a multi-criteria approach and possibly consider an ensemble of non-dominated, Pareto-efficient candidates.

- I think that the use of tools aimed at improving the explainability of machine learning algorithms is quite an interesting part of this contribution. However, I think it could be useful to have an expanded introduction about the differences existing among the various metrics and approaches. As an example, what is the difference between partial dependence, Friedman's H, and Shapley values? The distinction might be obvious to experts in machine learning, but quite obscure -- at least in its nouances -- to readers with a more traditional epidemiological background. 

- As for the Shapley plots, the random selection of a a small number (six) of locations seems to a bit arbitrary, and I wonder what would be the robustness of the results if/when other sites were to be selected. If this is feasible at all, I would recommend an exhaustive `beeswarm' analysis.

Minor comments
- l.24: <85% or >85% ?
- l.49: if those countries are disease-free, what is the cause of the economic losses? This should be explained
- l.98: most significant?
- l.144: "whether the prediction is correct or incorrect" unclear, what do you mean?
- l.164: "comprising 58.1%" I think you mean that 58.1% of cases involved cattle, but this should be better phrased
- l.178: the caption should be revised so that the descriptions actually match the panels they are associated with
- l.190: "These variables... climatic conditions" the main clause is missing
- l.229: these "background data" seem to be equivalent to pseudoabsences in `classical' species distribution models -- this should perhaps be mentioned
- l.230: "Since MFD... positive locations" This is quite confusing, please explain
- l.328: stable -> constant
- l.339: "Furthermore... 3.3 ms-1" there is something wrong/missing with the construction of this sentence, please rephrase
- l.348: I would suggest capitalizing the beginning of each sentence starting after a full stop
- l.376: A, D, I -> A, D, G
- l.381: same comment as l.348
- l.382: "had no specific unit" how so? Isn't it the same as temperature? And why does it need to be scaled by 100?
- l.387: same comment as l.348
- l.392: Conversely -> In contrast
- l.393: why is "Buffalo" capitalized?
- l.395: same comment as l.393
- l.492: same comment as l.328
- l.549: a randomly -> in a randomly

Author Response

Reviewer 1 1. The manuscript by Dr. Alkhamis and colleagues deals with a machine learning approach to associate foot-and-mouth disease cases with known environmental and eco-epidemiological drivers. The topic of this manuscript is quite interesting, due to the widespread impacts of foot-and-mouth disease in large geographical regions, especially in North Africa and the Middle East. The manuscript is well written and well organized. I only have a couple of technical points on the approach, and some minor comments -- all detailed below. - We thank the reviewer for their valuable time and suggestions for improving the present manuscript. Technical comments 2. I understand, in principle, the inclusion of multiple metrics to evaluate the performances of the machine learning algorithms. However, when it comes to selecting the best performing models, the presence of alternative metrics makes the choice more complicated. For instance, for the `All Serotypes' model, XGB is identified as the best candidate, yet it is outperformed by RF as far as specificity is concerned. As for the `Serotype O' model, RF is selected but XGB outperforms it in terms of accuracy. Trade-offs like these are to be somehow expected in the presence of multiple evaluation metrics, but what is the criterion used for model selection is never formally discussed here. Personally, I would recommend to pick just one metric to conduct the selection or -- perhaps even better -- to adopt a multi-criteria approach and possibly consider an ensemble of non-dominated, Pareto-efficient candidates.   -  While we agree with the reviewer’s comment, our model selection criteria are considered the gold standard for ML analytical statistical frameworks. We initially focused solely on accuracy and MCC values to select the best predictive model. However, for serotype O, the accuracy and MCC were identical for RF and XGB (see table 1), and therefore, our decision to select the RF model was based on its slight outperformance in the Sp and Se values. Thus, we followed a semi-hierarchical approach in our model selection process, which was emphasized in the text (see lines 385-389). Yet, we are currently working on developing an even better approach that averages the prediction over several ML algorithms implemented in the TidySDM by Leonardi et al., 2024  , instead of resorting to the model selection process in our current analytical pipeline. However, the current package does not allow us to estimate individual-level two-way interactions or shaply values, which a critical part of our analytical pipeline. We also highlighted this point in the limitation section of the discussion (see lines 1085- 1091).   3. I think that the use of tools aimed at improving the explainability of machine learning algorithms is quite an interesting part of this contribution. However, I think it could be useful to have an expanded introduction about the differences existing among the various metrics and approaches. As an example, what is the difference between partial dependence, Friedman's H, and Shapley values? The distinction might be obvious to experts in machine learning, but quite obscure -- at least in its nouances -- to readers with a more traditional epidemiological background.  - We briefly described the partial dependence, Friedman's H, and Shapley values, in the model interpretation section from the methods, as suggested by the reviewer (see lines 362-381). 4. As for the Shapley plots, the random selection of a a small number (six) of locations seems to a bit arbitrary, and I wonder what would be the robustness of the results if/when other sites were to be selected. If this is feasible at all, I would recommend an exhaustive `beeswarm' analysis. - It is important to note the beeswarm analysis suggested by the reviewer is analogous to  global Shapely value analysis. We actually explored this approach and it yielded similar results to the feature importance results in figures 3a,d, and g. Thus, it was excluded to avoid redundancies. In epidemiological terms, global Shapley values reveal the overall importance and magnitude of key features across the entire study area, whereas individual Shapley values expose the local ecological dynamics by showing where and why certain regions emerge as disease hotspots while others remain unaffected. We made this point clear in the last section of the discussion (see lines 1067-1073). Minor comments 5. l.24: <85% or >85% ? - line 25: We fixed the sign as recommended by the reviewer. 6. l.49: if those countries are disease-free, what is the cause of the economic losses? This should be explained - lines 53-54: We added a sentence to explain the cause of the economic losses in FMD-free countries as suggested by the reviewer. 7. l.98: most significant? - Line 104: Yes, correct, we used the word “most” as suggested by the reviewer. 8. l.144: "whether the prediction is correct or incorrect" unclear, what do you mean? - Lines 149: We removed the sentence to reduce the reader's confusion, as the reviewer suggested.  9. l.164: "comprising 58.1%" I think you mean that 58.1% of cases involved cattle, but this should be better phrased -lines 185-187: we rephrased the sentence as suggested by the reviewer. 10. l.178: the caption should be revised so that the descriptions actually match the panels they are associated with - lines 217-219: We fix the panel description order as the reviewer pointed out. 11. l.190: "These variables... climatic conditions" the main clause is missing - line 228: We fix the sentence as suggested by the reviewer. 12. l.229: these "background data" seem to be equivalent to pseudoabsences in `classical' species distribution models -- this should perhaps be mentioned - lines 291-296: We added a sentence with a reference to acknowledge the point raised by the reviewer. 13. l.230: "Since MFD... positive locations" This is quite confusing, please explain Lines 295-296: We fixed and clarified the sentences as suggested by the reviewer. 14. l.328: stable -> constant - Line 452: We fixed the word as suggested by the reviewer. 15. l.339: "Furthermore... 3.3 ms-1" there is something wrong/missing with the construction of this sentence, please rephrase - Lines 463-466: We fixed the sentence and the superscript for the unit of measurement for wind speed, which is in meters per second (ms-1), as suggested by the reviewer. 16. l.348: I would suggest capitalizing the beginning of each sentence starting after a full stop - We capitalized all of the figures as suggested by the reviewer. 17. l.376: A, D, I -> A, D, G - Fixed as suggested by the reviewer. 18. l.381: same comment as l.348 - We capitalized all of the figures as suggested by the reviewer. 19. l.382: "had no specific unit" how so? Isn't it the same as temperature? And why does it need to be scaled by 100? - In climatology, temperature seasonality (BIO4) originates as the standard deviation of monthly mean temperatures in °C, but in WorldClim it is scaled by 100 for storage efficiency and consistency across variables. While mathematically derived in °C, after scaling it is treated as a dimensionless index, used comparatively to indicate the degree of seasonal variability rather than a direct temperature measure. 20. l.387: same comment as l.348 - We capitalized all of the figures as suggested by the reviewer. 21. l.392: Conversely -> In contrast - We fixed the word as suggested by the reviewer. 22. l.393: why is "Buffalo" capitalized? - We fixed the word throughout the manuscript as the reviewer suggested. 23. l.395: same comment as l.393 - We fixed the word throughout the manuscript as the reviewer suggested. 24. l.492: same comment as l.328 - Line 554: We fixed the word as suggested by the reviewer. 25. l.549: a randomly -> in a randomly - Line 554: We fixed the word as suggested by the reviewer.

Reviewer 2 Report

Comments and Suggestions for Authors

Foot-and-mouth disease (FMD) remains a devastating threat to livestock health and food security in the Middle East and North Africa (MENA), where complex interactions among host, environmental, and anthropogenic factors constitute an optimal endemic landscape for virus circulation. Further implementation of our analytical pipeline to guide risk-based surveillance programs and intervention efforts will help reduce the economic and public health impacts of this devastating animal pathogen. However, there are some issues with the manuscript that need to be addressed urgently.

  1. Section 2: Materials and Methods contains no explicit statements. Is this appropriate? We recommend the authors consider this.
  2. It is recommended that each figure in the diagram be labeled with a lowercase letter.For example, in Figure 1, it is recommended to use a, b, c, and d.
  3. For Figures 2 to 5, would it be more appropriate to explain the text following the bolded sections within the main text? Otherwise, the lengthy issues following the figures indicate that the authors' illustrations may be difficult to interpret clearly and require further refinement.
  4. Chapter 4. Discussion is too lengthy. It is recommended that the author break it down into bullet points rather than presenting so many paragraphs at once, as this makes the structure unclear.

Author Response

Reviewer 2 1. Foot-and-mouth disease (FMD) remains a devastating threat to livestock health and food security in the Middle East and North Africa (MENA), where complex interactions among host, environmental, and anthropogenic factors constitute an optimal endemic landscape for virus circulation. Further implementation of our analytical pipeline to guide risk-based surveillance programs and intervention efforts will help reduce the economic and public health impacts of this devastating animal pathogen. However, there are some issues with the manuscript that need to be addressed urgently. - We thank the reviewer for their valuable time and suggestions for improving the present manuscript. 2. Section 2: Materials and Methods contains no explicit statements. Is this appropriate? We recommend the authors consider this. - We added an explicit statement at the beginning of the methods section as suggested by the reviewer. 3. It is recommended that each figure in the diagram be labeled with a lowercase letter.For example, in Figure 1, it is recommended to use a, b, c, and d. - We fixed the figure labels to lowercase throughout the manuscript as the reviewer suggested. 4. For Figures 2 to 5, would it be more appropriate to explain the text following the bolded sections within the main text? Otherwise, the lengthy issues following the figures indicate that the authors' illustrations may be difficult to interpret clearly and require further refinement. - Text after the bold figure caption is simply a description of the figure elements, not an interpretation of results. Moving this text from the figure to the main text could cause confusion, as seen in Figure 2, where the main legend lists the figure panels, explains what the dots on the maps represent, and describes the values of the color gradient on the heat map. Therefore, it is unusual to remove such sentences from the figure captions to the main text. See similar research papers: 1. https://besjournals.onlinelibrary.wiley.com/doi/full/10.1111/1365-2656.13076/ 2. https://esajournals.onlinelibrary.wiley.com/doi/10.1002/eap.2407   5. Chapter 4. Discussion is too lengthy. It is recommended that the author break it down into bullet points rather than presenting so many paragraphs at once, as this makes the structure unclear. - We divided the discussion into sub-headed sections as recommended by the reviewer.

Reviewer 3 Report

Comments and Suggestions for Authors

Author Response

Reviewer 3 In this study, the authors applied an interpretable machine learning (ML) statistical framework to model the epidemiological landscape of FMD between 2005 and 2025. Furthermore, they compared the ecological niche of serotypes O and A in the MENA region. This paper is well organized and the derived conclusions own important meaning in biology. - We thank the reviewer for their valuable time and suggestions for improving the present manuscript. 1. The key contribution of this paper shall be summarized in Introduction. - Lines: 152-157: We add a paragraph about the key contribution of this paper as suggested by the reviewer. 2. The structure of this paper shall be given in Introduction. - Lines: 161-171: We provided a summary for the structure of the paper as a statement in the first part of the methods section as suggested by the reviewer. 3. In Section 2.3, what is the merit of data processing in this paper? The data processing in this paper carries several important merits that strengthen the validity and interpretability of the modeling framework: 1. Integration of heterogeneous data sources: The study harmonized outbreak, environmental, host, and anthropogenic datasets from multiple international databases (FAO EMPRES-i, WOAH-WAHIS, WorldClim, FAO GeoNetwork, etc.), ensuring a comprehensive representation of ecological and epidemiological factors influencing FMD dynamics across the MENA region. 2. Standardization and spatial alignment: All raster layers were resampled and projected to a uniform 10 km² spatial grid, which minimized spatial bias and enabled consistent, high-resolution comparisons across diverse data types. 3. Reduction of collinearity and noise: The use of correlation filtering (|ρ| > 0.7) and the Boruta feature selection algorithm eliminated redundant or non-informative predictors, improving model efficiency, stability, and interpretability. 4. Generation of pseudo-absence (background) data: Since true absence data are rarely available for transboundary diseases like FMD, the study employed a balanced presence-absence framework using carefully generated background points to mimic ecological variability and reduce sampling bias. 5. Rigorous validation design: By adopting an 80/20 train–test split and K-fold cross-validation, the data processing pipeline ensured reliable model generalization and prevented overfitting. We summarized this point in the last section of the discussion, see lines 1001-1006. 4. In Section 2.4, what is the advantage of the proposed ML algorithms? - Lines 323-331: we added a few sentences in that regard as suggested by the reviewer. 5. Compared with the previous works, what is the advantage/disadvantage of the proposed ML statistical framework? - See sections 4.4 and 4.5 for detailed disadvantageadvantages of our proposed ML pipelines. 6. The format of the references shall be united. - We revised and unified the format of all references as suggested by the reviewer.

Round 2

Reviewer 2 Report

Comments and Suggestions for Authors

The conclusion section could be further enriched. We recommend refining and optimizing it.

Author Response

Comment 1: The conclusion section could be further enriched. We recommend refining and optimizing it.

Response 1: The conclusion section has been enriched and refined as suggested by the reviewer